# Evaluation of a successful fluoroquinolone restriction intervention among high-risk patients: A mixed-methods study

Jessica Tischendorf[1]*, Matthew Brunner[1], Mary Jo Knobloch[1,2], Lucas Schulz[3], Anna Barker[4,5], Marc-Oliver Wright[6], Alexander Lepak[1], Nasia Safdar[1,2]

1 Department of Medicine, University of Wisconsin School of Medicine and Public Health, Madison, Wisconsin, United States of America, 2 William S. Middleton Memorial Veterans Hospital, Madison, Wisconsin, United States of America, 3 Department of Pharmacy, University of Wisconsin Hospital and Clinics, Madison, Wisconsin, United States of America, 4 Department of Population Health Sciences, University of Wisconsin Madison, Madison, Wisconsin, United States of America, 5 Department of Internal Medicine, University of Michigan Ann Arbor, Ann Arbor, Michigan, United States of America, 6 Infection Prevention Program, University of Wisconsin Hospital and Clinics, Madison, Wisconsin, United States of America

* jtischendorf@wisc.edu

**Data Availability Statement:** All relevant data are within the manuscript and supporting information files.

## Abstract

### Objective

We conducted a quality improvement initiative to restrict fluoroquinolone prescribing on two inpatient units housing high-risk patients and applied a human factors approach to understanding the barriers and facilitators to success of this intervention by front-line providers.

### Methods

This was a mixed-methods, quasi-experimental study. This study was conducted on two inpatient units at a tertiary care academic medical center: the medical-surgical intensive care and abdominal solid organ transplant units. Unit-level data were collected retrospectively for 24 months pre- and post- fluoroquinolone restriction intervention, implemented in July 2016, for all admissions to the study units. Our restriction intervention required antimicrobial stewardship pre-approval for fluoroquinolone prescribing. We explored barriers and facilitators to optimal fluoroquinolone prescribing using semi-structured interviews attending, fellow and resident physicians, advanced practice providers and pharmacists on these units.

### Results

Hospital-onset *C. difficile* infection did not decrease significantly, but fluoroquinolone use declined significantly from 111.6 to 19.8 days of therapy per 1000 patient-days without negatively impacting length of stay, readmissions or mortality. Third generation cephalosporin and aminoglycoside use increased post-restriction. Providers identified our institution's strong antimicrobial stewardship program and pharmacy involvement in antimicrobial

**Funding:** Nasia Safdar is supported by grant number R01HS026226 from the Agency for Healthcare Research and Quality. The content is solely the responsibility of the authors and does not necessarily represent the official views of the Agency for Healthcare Research and Quality.

**Competing interests:** No authors have competing interests.

decision making as key facilitators of fluoroquinolone optimization and patient complexity, lack of provider education and organizational culture as barriers to optimal prescribing.

## Conclusions

Fluoroquinolones can be safely restricted even among high-risk patients without negatively impacting length of stay, readmissions or mortality. Our study provides a framework for successful antimicrobial stewardship interventions informed by perceptions of front line providers.

## Introduction

*Clostridioides difficile* infection (CDI) is the leading cause of healthcare-associated diarrhea and accounts for 12% of all hospital-acquired infections (HAIs) in the United States (US) [1,2]. Antibiotic use is the major driver of healthcare-associated CDI [3,4], causing gut microbiome disruption and proliferation and propagation of *C. difficile*. The risk of CDI differs by antibiotic class. Fluoroquinolones have been shown to greatly increase the risk of CDI [5–8].

Antibiotic stewardship programs (ASPs) can reduce CDI incidence [8–13] through governance of high-risk antimicrobials, with restriction policies being particularly efficacious [8,9,14–20]. Fluoroquinolone restriction to reduce CDI rates has been successfully applied during outbreaks, [15,20,21] though the impact in the endemic setting has not been fully described. Further, the data on fluoroquinolone restriction among special populations, such as the immunocompromised and critically ill, are limited.

With any antibiotic stewardship intervention, attention to implementation is critical to assess the impact on prescriber behavior, workflow and patient and institutional outcomes [11]. Understanding not just patient and provider-level influences on prescribing, but also work system factors, can inform a successful, sustainable intervention to optimize prescribing.

We undertook a quality improvement initiative to implement a fluoroquinolone restriction intervention in the intensive care unit (ICU) and solid organ transplant (SOT) unit to attempt to decrease our hospital-onset CDI (HO-CDI) rate. We interviewed front-line providers on the pilot units to understand the work system factors influencing fluoroquinolone prescribing. By applying human factors and ergonomics concepts, we sought to identify facilitators and barriers to a successful restriction intervention and apply these factors to larger scale antimicrobial stewardship interventions.

## Methods

### Setting

University of Wisconsin (UW) Hospital is a 592-bed tertiary care academic center in Madison, Wisconsin. Our fluoroquinolone restriction was instituted on two units housing patients at high infectious risk: a 24-bed medical-surgical ICU and a 32-bed medical-surgical abdominal SOT ward. UW Hospital performs more than 250 renal transplants and more than 100 liver transplants annually.

## Study design

This was a mixed-methods quasi-experimental study. Pre-intervention unit-level data were collected for a 24-month period before the intervention (July 2014 to June 2016), and for a 24-month period after the intervention (July 2016 to June 2018).

## Outcomes

The primary quantitative outcome was rate of HO-CDI. CDI cases were defined as positive laboratory test result for *C. difficile* on unformed stool in a patient with diarrhea using the Centers for Disease Control and Prevention's National Healthcare Safety Network criteria [22]. We included all positive tests regardless of time from admission. Our institution uses the *Clostridioides (Clostridium) difficile* toxin B PCR (Cepheid©, Sunnyvale, California).

Secondary outcomes were fluoroquinolone and alternative antimicrobial days of therapy (DOT), length of stay (LOS), readmissions and mortality. We measured antimicrobial use as days of therapy (DOT) per 1,000 patient days. To assess potential harms, we measured LOS, readmissions and in-hospital mortality for the pilot units. LOS was measured in days and evaluated for both the study units and total index hospitalization. Readmissions were measured as percent of patients admitted to the study units any time during the index admission who were readmitted to any unit within 30 days of discharge. In-hospital mortality was assessed.

## Intervention

In July 2016, the UW Antimicrobial Stewardship Program (ASP) instituted a fluoroquinolone restriction intervention on two units. UW Hospital has a well-established restricted antimicrobial formulary, which did not include fluoroquinolones prior to 2016. For restricted antimicrobials, pre-authorization by an ASP physician or pharmacist prior to electronic order entry is required. Necessary pre-authorization is obtained by ordering providers through an ASP pager staffed by an infectious disease physician or pharmacist between the hours of 7:00 AM to 11:00 PM seven days per week. After hours, a conditional dose can be administered at the discretion of primary providers. Our ASP also performs daily prospective audit and review of all inpatients on antimicrobials.

The fluoroquinolone restriction on study units was implemented in a manner similar to other restricted antimicrobials as described above. Operationally, when a provider ordered a fluoroquinolone, a hard stop in the electronic order entry process appeared indicating fluoroquinolones were restricted and required ASP approval prior to ordering. Several exemptions to the restriction were agreed upon by the ASP, including: recommendation by infectious disease consultation, chemotherapy-induced neutropenia requiring prophylaxis, periprocedural use for urological or select surgical procedures in patients with immediate IgE-mediated β-lactam allergy, and cystic fibrosis exacerbations. Selections for exempted conditions were available at the time of order entry, allowing the provider to continue through the electronic ordering process without seeking ASP approval if exemption criteria were met. If no exemption criteria were met, providers were required to contact the ASP pager for approval, and if approved, indicate the approving provider in the electronic order. If a patient was transferred to one of the intervention units with an active fluoroquinolone prescription, the prescription required re-authorization from the ASP in the usual manner. On other hospital units, fluoroquinolones could be used at provider discretion without prior approval though were subject to routine daily prospective audit and feedback in line with other antimicrobial use. Discharge prescriptions for fluoroquinolones were not monitored or restricted.

Education and clinical decision support tools were provided to pharmacists and physicians on study units regarding rationale for restriction and fluoroquinolone alternatives. Alternative,

non-fluoroquinolone treatment options for common infections were posted online (available at www.uwhealth.com/cckm). Fluoroquinolone alternatives tables and order panels to assist decision making were available in the EMR (Epic©, Verona WI). Clinical pharmacists are present 24 hours per day in the ICU and 16 hours per day on the SOT unit for consultation.

## Concurrent infection prevention interventions

Throughout the 48-month study period, multiple concurrent and overlapping infection prevention interventions were implemented to reduce HO-CDI (Table 1). Briefly, interventions were implemented at all levels: screening select patients on admission, optimizing appropriateness of CDI testing, covert isolation and hand hygiene compliance monitoring and feedback and enhanced environmental decontamination procedures.

## Pharmacy review for adherence and harm

For one month following implementation of the fluoroquinolone restriction, pharmacy services prospectively collected data on all patients who received antimicrobials on the pilot units. Indications selected by providers through electronic order entry were collected. Rate of AKI after antimicrobial initiation were assessed. Adherence to the restriction policy was assessed, defined as the selection of an appropriate alternative agent based on institutional guidance available to prescribers as part of the education campaign. AKI was defined by the 2012 Kidney Disease Improving Global Outcomes guidelines [23].

## Unit level data

Our data analytics team provided unit-level data for 24 months pre- and 24 months post-intervention, including LOS, readmissions, deaths, and HO- CDI cases. Antimicrobial prescribing data were extracted from the EMR by our data analytics team and was validated iteratively by investigators (M.B. and J.T.).

**Table 1. Concurrent infection prevention interventions to reduce CDI at UW Hospital.**

| | Month-Year | Study Design Period(s) Impacted |
|---|---|---|
| Expanded duration of isolation precautions from 30 to 90 days from most recent positive | Oct-15 | I (partial)and II (all) |
| Implementation of CDI testing algorithm for all inpatients | Dec-15 | I (partial)and II (all) |
| Admission screening of bone marrow transplant patients on one of the pilot units | Dec-15 | I (partial)and II (all) |
| Covert isolation and hand hygiene compliance observations and feedback to nursing unit and hospital leadership | Dec-15 | I (partial)and II (all) |
| Electronic medical record alert to test symptomatic patients on admission | Jun-16 | I (partial)and II (all) |
| Ultraviolet light cleaning | Sep-16 | II |
| Electronic medical record alert *not* to test asymptomatic patients or those receiving laxatives | Oct-16 | II |
| Pre-existing fluorescent marking system of high touch objects (HTO) was doubled from the 16 CDC and vendor recommended high touch objects to 32. | Dec-16 | II |
| Established antimicrobial stewardship program staffed by infectious disease physicians and pharmacists performing daily prospective audit and feedback and front-line restriction of several antimicrobials. | | I (all) and II (all) |

## Provider interviews and conceptual framework

We conducted semi-structured interviews of providers on the pilot units, including attending physicians, fellows, residents, advanced practice providers and pharmacists. Interviews were conducted until emergent themes were clearly identified. Interviews were conducted following the implementation of the restriction intervention. We used the Systems Engineering Initiative for Patient Safety (SEIPS) framework to develop our interview guide (online supplement) and as a framework to examine the process of antibiotic decision-making. This model depicts a work system defined by the interaction of five elements: the *individual* who performs the *tasks* using *tools* in a physical *environment* and within an *organizational* infrastructure. Research related to SEIPS examines job and systems design, quality improvement, and technology implementation that affect patient safety outcomes related to patients, organizations and staff [24]. The SEIPS model emphasizes the interactions among work systems components, recognizing that all changes in a given work system influence the remaining elements. This model is well suited to describe influences on the complex decision making behind antimicrobial prescribing.

In our provider interviews, we focused on (1) perception of fluoroquinolone utility, (2) indications used for fluoroquinolone use, (3) perception of the relationship between fluoroquinolone use and CDI, and (4) barriers to an intervention successfully restricting fluoroquinolone use. Three pilot interviews were conducted after which the interview guide was refined. Interviews were conducted by two trained investigators after a joint interview facilitated concordance in question-asking. Interviews were audio-recorded and transcribed verbatim. Transcripts were coded by two investigators using a deductive method with components of SEIPS as predetermined themes.

## Data management and statistical analysis

Pairwise comparisons between pre and post-intervention mean estimates of all primary and secondary outcomes was performed using the two-sample t-test. An assessment of normality was performed for continuous variables using qq-plots. Five-point moving averages were used to smooth the HO-CDI data. Interrupted time series analysis was subsequently conducted using Prais-Winsten regression to account for first-order autocorrelation between measurements. All statistical analyses were conducted in STATA using an alpha significance level of ≤0.05.

Qualitative data was coded into categories within the SEIPS framework (people, organization, tools/technology, tasks, and environment) using DeDoose© (SocioCultural Research Consultants, LLC, Manhattan Beach, California). Emergent themes within each SEIPS component were identified and categorized as a facilitator or barrier to optimizing fluoroquinolone prescribing. DeDoose© provides the opportunity to analyze frequency of code co-occurrence (the number of times the researchers coded constructs together). This information allowed the research team to identify themes coded together for the same passage of text.

## Ethics and reporting

This was a quality improvement initiative and was deemed exempt from institutional review board oversight. We followed SQUIRE 2.0 guidelines in the reporting of our quality improvement study [25]. Verbal consent was obtained from interviewees prior to semi-structured interviews. Qualitative analysis was conducted in a de-identified manner, preventing attribution of statements to any particular interviewee.

## Results

The average HO-CDI rate in the study units was lower in the post-intervention period (11.8 vs 22.2 infections per 10,000 patient days, p = 0.001). However, time series analysis (Table 2, Fig 1) showed no significant change in HO-CDI rate at the time of intervention implementation (reduction of 3.4 infections/10,000 PD, p = 0.20). Time series analysis demonstrated an increase in the trend of the infection rate in the post-intervention period (increase of 0.8 infections/10,000 patient days, per month, p = 0.002).

In our combined medical-surgical ICU and abdominal SOT ward, fluoroquinolones are used most commonly for pneumonia, intraabdominal infections and urinary tract infections (Table 3). Fluoroquinolone use decreased from an average of 111.6 DOT/1000 patient-days in the pre-intervention period to 19.8 DOT/1000 patient-days post-intervention (p<0.001). Total antimicrobial days of therapy in the pre-intervention period was 1664 DOT/1000 patient-days and in the post-intervention period, 1543 DOT/1000 patient-days. The average readmission rate, LOS on intervention units, and use of fourth generation cephalosporins decreased post-intervention. In contrast, use of third generation cephalosporins, aminoglycosides, and piperacillin-tazobactam increased post-intervention (Table 4).

### Pharmacy services review

In the one-month period following implementation of the fluoroquinolone restriction intervention, 138 antimicrobial treatment courses were prescribed to 129 patients on the pilot units. Indications for all antimicrobial prescribing for this time period are listed in S1 Table, with the most common being urinary tract infection on the SOT unit and sepsis or septic shock in the ICU. Among the 129 patients, 40 (31%) had an antimicrobial allergy label in their chart. Twenty-two of these 40 (55%) patients had beta-lactam allergy. Overall adherence to the fluoroquinolone restriction policy was 84.6%, with syndrome specific policy adherence indicated in S1 Table. Six patients developed AKI (11.5%).

### Provider interviews

We conducted twelve interviews among residents, fellows, attending physicians, advanced practice providers and pharmacists. The person component of the work system was discussed most frequently, with person as facilitator ranking the highest (52 times coded) and person as

**Table 2. Patient outcomes and antimicrobial utilization.**

|  | Pre | Post | p-value |
|---|---|---|---|
| Readmission rate (%) | 18.7 | 17.3 | 0.028 |
| Length of stay, total encounter (days) | 8.6 | 8.4 | 0.20 |
| Length of stay on study units (days) | 5.0 | 4.7 | 0.009 |
| In-hospital mortality (%) | 7.7 | 8.4 | 0.10 |
| HO-CDI per 10,000 patient-days | 22.2 | 11.8 | 0.001 |
| Fluoroquinolones[a] | 111.6 | 19.8 | <0.001 |
| Carbapenems[a] | 40.4 | 46.3 | 0.32 |
| Third Generation Cephalosporins[a] | 88.1 | 109.0 | <0.001 |
| Aminoglycosides[a] | 3.2 | 4.8 | 0.04 |
| 4th Generation Cephalosporins[a] | 109.1 | 37.6 | <0.001 |
| Piperacillin/Tazobactam[a] | 134.8 | 199.6 | 0.001 |

[a]All antibiotics are reported in days of therapy per 1,000 patient days

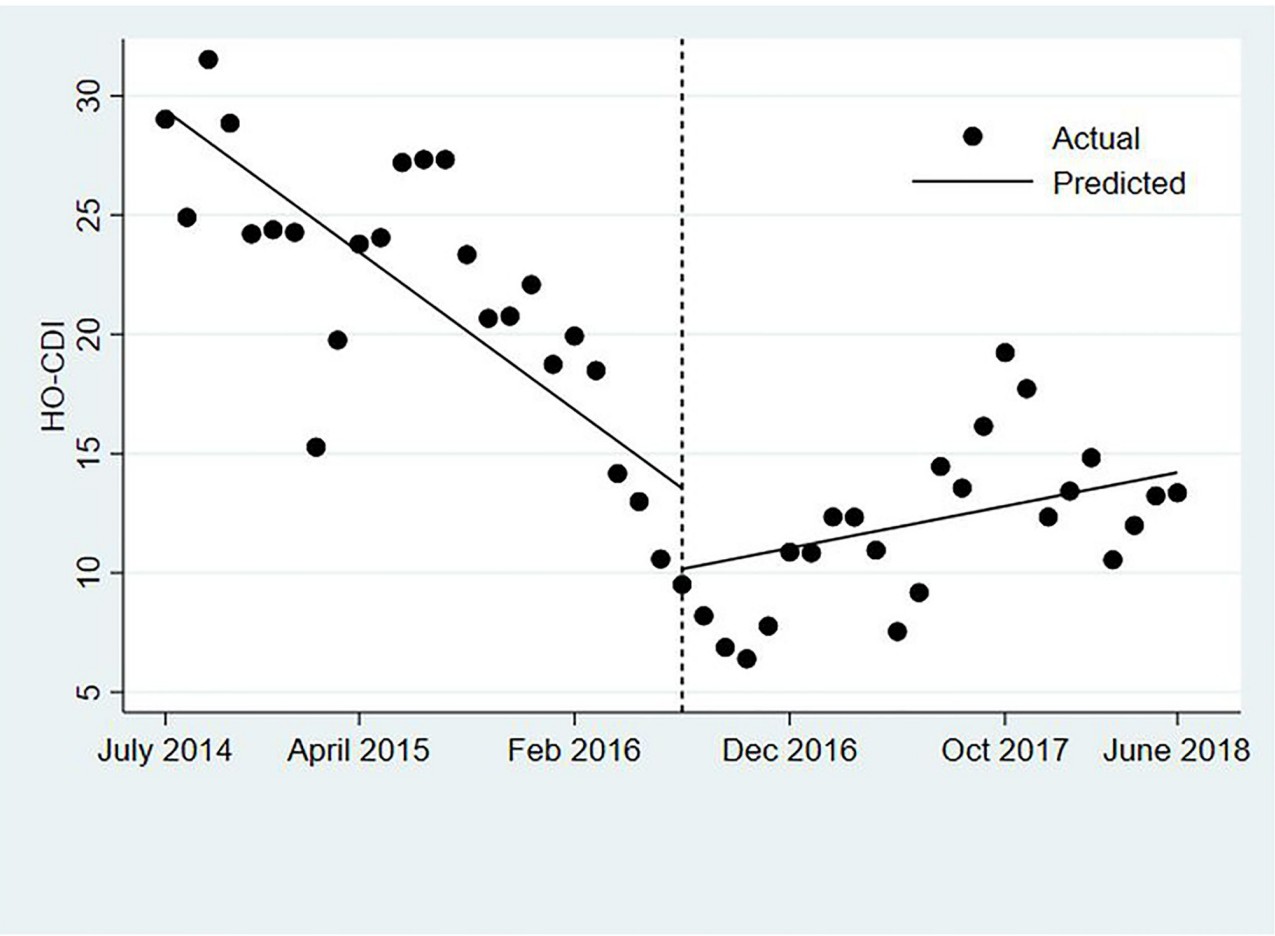

**Fig 1. Interrupted time-series analysis, hospital-onset *Clostridioides difficile* infection.**

barrier ranking second highest (41 times coded). The next most frequently coded SEIPS components were organization as facilitator (39 times coded) and tools and technology as facilitator (38 times coded). Organization as a barrier was coded 23 times. Person and facilitator codes were used together 8 times during the interviews. The second most frequently co-occurring codes were organization and facilitator (7 times). However, the categories of persons, organizations and tasks were co-coded as barriers almost as often (6 times). The category of environment was co-coded with barrier five times. We have highlighted emergent themes, with the most common barriers and facilitators to optimizing fluoroquinolone prescribing with epitomizing quotes in Table 5. Patient complexity, lack of provider education, and

**Table 3. Time series analysis hospital onset-*Clostridioides* difficile infection.**

| Factor | Intervention units | |
|---|---|---|
| | Coefficient | p-value |
| Intercept (HO-CDI per 10,000 patient days) | 29.4 | <0.001 |
| Slope pre-intervention (change in HO-CDI per 10,000 patient days per month) | -0.7 | <0.001 |
| Immediate effect at time of intervention (HO-CDI per 10,000 patient days) | -3.4 | 0.20 |
| Difference between pre- and post-intervention slopes (HO-CDI per 10,000 patient days per month) | 0.8 | 0.002 |

**Table 4. Most common indications for fluoroquinolone use on pilot units.**

| | Medical-Surgical ICU | | Abdominal SOT ward | |
|---|---|---|---|---|
| | Pre-intervention | Post-intervention | Pre-intervention | Post-intervention |
| Lower respiratory tract infection/Pneumonia | 43.6% | 44.9% | 19.4% | 18.0% |
| Abdominal/Pelvic | 19.8% | 28.6% | 32.5% | 38.0% |
| Bloodstream | 14.7% | 0.0% | 5.3% | 8.0% |
| Urinary tract infection | 10.1% | 8.2% | 21.9% | 26.0% |
| Upper respiratory tract infection | 5.8% | 2.0% | 4.6% | 0.0% |
| Cellulitis | 1.9% | 12.2% | 2.1% | 0.0% |
| Meningitis | 0.5% | 0.0% | 0.0% | 0.0% |
| Surgical Wound | 1.0% | 2.0% | 3.9% | 2.0% |
| Head and Neck infection | 0.5% | 0.0% | 1.1% | 0.0% |
| Musculoskeletal/Osteomyelitis | 0.5% | 0.0% | 0.4% | 2.0% |
| Transplanted Organ | 0.7% | 0.0% | 8.5% | 6.0% |
| Neutropenic Fever | 0.0% | 2.0% | 0.0% | 0.0% |
| Non-infectious | 0.0% | 0.0% | 0.4% | 0.0% |
| Intravenous Line | 0.0% | 0.0% | 0.0% | 0.0% |
| Other | 0.0% | 0.0% | 0.0% | 0.0% |

All indications expressed as percentage of total fluoroquinolone days of therapy.

ICU: intensive care unit; SOT: solid organ transplant

organizational culture were identified as barriers. Strength of the ASP and pharmacy involvement in antimicrobial decision making were identified as key facilitators of successful fluoroquinolone restriction (Fig 2).

## Discussion

An antimicrobial stewardship intervention to restrict prescribing of fluoroquinolones can lead to substantial decrease in their use, even among high-risk patients. While we did not observe a reduction in HO-CDI following our intervention, we did demonstrate fluoroquinolones could be restricted among critically ill and immunocompromised patients without negatively impacting LOS, readmission rate and mortality on our pilot units. Following restriction of fluoroquinolones, we did observe an increase in third-generation cephalosporin and aminoglycoside prescribing, though overall antimicrobial days of therapy did not increase. Fortunately, despite the increase in aminoglycoside use we did not observe a high rate of in AKI. Indications for prescribing in our population are similar to mirror those in another successful fluoroquinolone restriction [26], with respiratory, urine and abdominal infections being the most common indications, though our focus on high-risk patients is unique.

One potential explanation for lack of HO-CDI reduction is that our pilot study was conducted in the endemic setting. In contrast to our study, in previous observational studies in the endemic setting, fluoroquinolone restriction alone [16,26] or as part of a campaign to restrict other antibiotics [17–19,27] has reduced HO-CDI. Like our study, many of these studies are limited by lack of control for patient-specific factors, with the exception of two in the outbreak setting [15,20]: one of which failed to demonstrate clear relationship between fluoroquinolone use and HO-CDI after multivariate analysis [15]. Studies in the epidemic setting demonstrated reduction in HO-CDI with fluoroquinolone restriction, especially among centers with a high burden of fluoroquinolone-resistant *C. difficile* [15,20,21]. In both the epidemic and endemic setting, fluoroquinolone restriction had particular impact on fluoroquinolone-resistant

**Table 5. Barriers and facilitators to fluoroquinolone prescribing optimization, coded by Systems Engineering Initiative for Patient Safety (SEIPS) framework component with illustrating provider quotes.**

| Quote | Element identified | SEIPS component |
|---|---|---|
| "Oftentimes it's difficult to . . . focus on every detail on, for every patient, especially when . . . the antibiotics that they're on is kind of the least of their issues." "They're [fluoroquinolones] good for troubled kidneys and the urinary tract infections and bacteremia" "So there were still perceptions and beliefs that double coverage of pseudomonas empirically was necessary kind of ignorant of patients' past culture history and local antibiogram" | *Barriers* Patient factors: complexity of acute illnesses and underlying comorbidities Provider belief about necessity to "double cover" for *Pseudomonas aeruginosa* | Person |
| "I think that it's exceedingly helpful to have our pharmacists here because they do it every day." | *Facilitator* Close involvement of pharmacists in patient care | Person |
| ". . .antibiotics are an area that's really complicated. It takes a lot of time and experience to learn well." | *Barrier* Lack of time to learn alternative regimens well | Task |
| ". . .they get broad coverage and full workup. And then generally the workup is negative and the antibiotics are deescalated, you know." | *Facilitator* Culture-directed de-escalation of empiric therapy | Task |
| INTERVIEWER: "How frequently do you see providers consulting the antibiogram for B4/6? RESPONDENT: "What antibiogram?" | *Barrier* Lack of awareness of antibiogram/poor visibility of antibiogram | Tools/ Technology |
| "There are rounding checklists in the ICU, and on that checklist is cultures and antibiotics intended to bring up the discussion of narrowing. And that certainly worked in like Foley and line usage. . . . I think the same concept can easily be applied and is frequently, not as consistently as lines and drains, but can be applied with antibiotics." | *Facilitator* Standardized rounding checklists | Tools/ Technology |
| ". . .for my experience the last five years I've been here, very much kind of a go-to class of antibiotics that I definitely think that we've used very frequently." | *Barrier* Optimizing fluoroquinolone use not perceived as organizational priority | Organization |
| "I know those weren't exactly cheap to run all the sensitivities and all their isolates. But I think . . . as an institution, that's an important place to invest, because we have to do our own local stewardship. And, you know, I think we've done a really good job of that. We don't have these pan-resistant organisms that other tertiary . . . centers are seeing." | *Facilitator* Strong antimicrobial stewardship presence | Organization |
| ". . .afternoon or evening or overnight from the emergency department, the resident probably comes up with the initial regimen." | *Barrier* Lack of strong pharmacy support in one unit overnight, when many empiric regimens are selected for new admissions | Environment |
| "Pharmacists are available 24/7 in TLC." | *Facilitator* Constant access to pharmacists in the ICU | Environment |

B4/6: abdominal solid organ transplant unit

ICU: intensive care unit

TLC: Trauma and Life Center, the name of UW Hospital intensive care unit

*C. difficile* [16,19,27]. We did not measure the rate of fluoroquinolone-resistant *C. difficile;* if low in this endemic setting, this may account for the lack of reduction in HO-CDI through restriction. The multiple, concurrent infection prevention interventions may have influenced the decreasing trend in HO-CDI observed on the pilot units prior to implementation of the fluoroquinolone restriction.

Another possible explanation for our intervention's lack of clear impact on HO-CDI rates is the rise in use of third generation cephalosporins and aminoglycosides, though fortunately without an increase in antimicrobial days of therapy overall. Other centers have seen increased use of alternative antimicrobials following fluoroquinolone restriction, specifically anti-pseudomonal penicillins [17], azithromycin [15], and aztreonam [15]. Others report no difference in antimicrobial consumption [16,19,26]. Given the frequency of fluoroquinolone use for pneumonia in our sample, we are not surprised by a resultant rise in ceftriaxone, an established

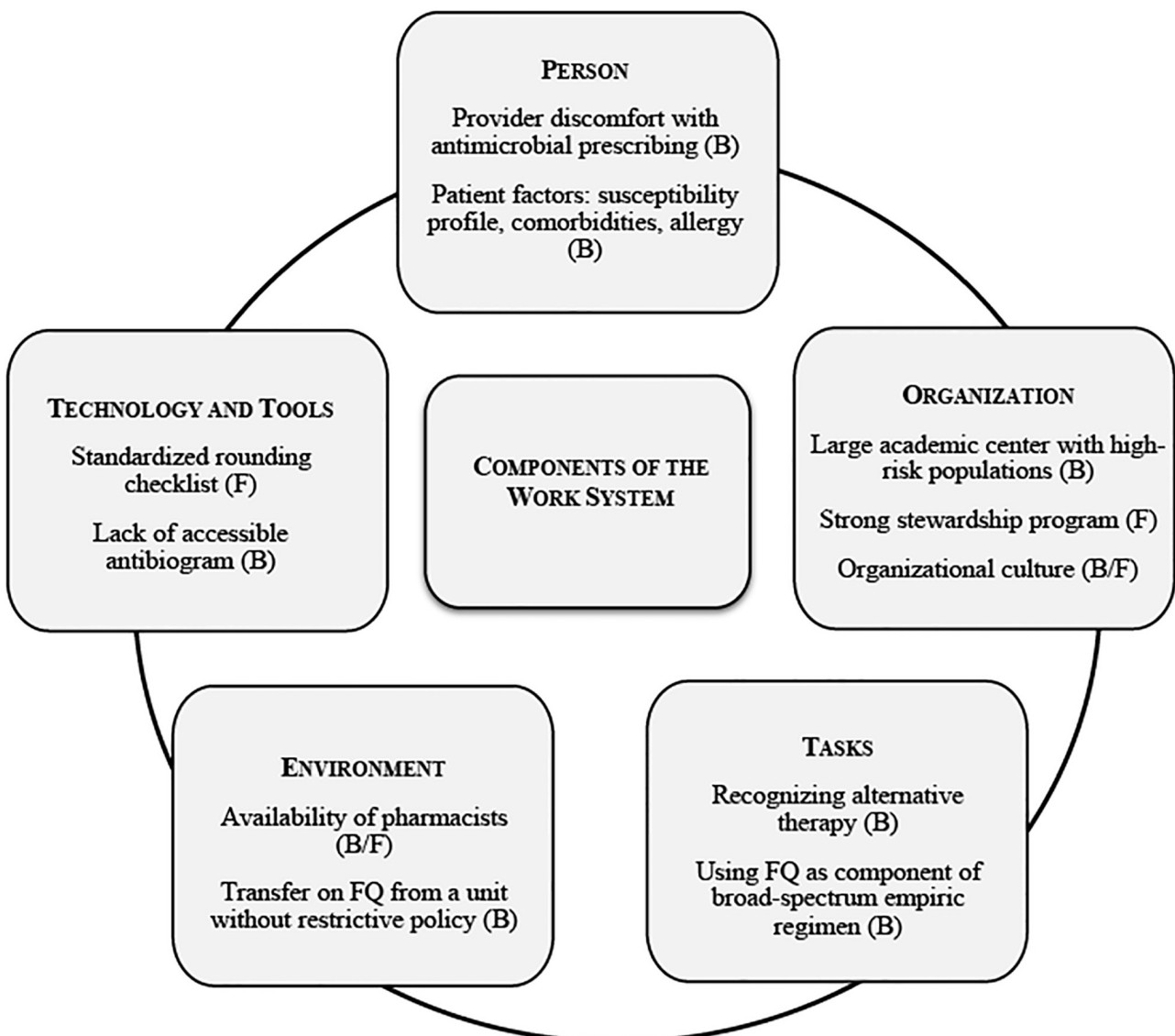

**Fig 2. Components of the work system influencing fluoroquinlone prescribing.** Components of the work system identified as barriers (B) or facilitators (F) to optimizing fluoroquinolone use by provider interviews on pilot units.

beta lactam treatment alternative. In our institution at the time, there was a strong culture of providing two empiric gram-negative agents to patients with sepsis, particularly among patients in the ICU, which likely explains the increase in aminoglycoside therapy following fluoroquinolone restriction. This phenomenon of "squeezing the balloon", that is, limiting availability of an agent resulting in an increase in use of other agents is an important unintended consequence to monitor [28]. Fortunately, despite more frequent aminoglycoside use there was not a high rate of AKI post-intervention. While we included fourth generation cephalosporins and piperacillin-tazobactam in our analysis, sequential drug shortages within our study period probably explain why utilization of these agents changed in opposite directions. It is interesting to note the high frequency of antimicrobial allergy among patients receiving fluoroquinolones, suggesting a need to better elucidate antimicrobial allergies to optimize prescribing.

Our interviewees identified the strength of our institution's ASP and the involvement of clinical pharmacists as key facilitators to the success of the fluoroquinolone restriction intervention. Clearly defining and dedicating resources to the ASP is necessary for success [29,30]. The importance of pharmacy involvement in ASPs as well as assistance with antimicrobial optimization at the patient level has been well described [30,33,37,31]. Key facilitators of success for our institution's ASP initiatives as well as those from the literature has informed a larger scale restriction intervention.

Front line providers identified several barriers to successfully implementing a fluoroquinolone restriction intervention. Patient complexity can complicate antimicrobial prescribing [29], which is certainly relevant in our cohort of critically ill and immunocompromised patients in whom the restriction intervention was piloted. A lack of provider education regarding appropriateness and fluoroquinolone alternatives was emphasized. Lack of education has been identified in previous qualitative work as a barrier to successful ASP interventions [31–33]. Similarly, others have identified provider education as a facilitator of success [29,34,35], either provided during the process of preauthorization or through more dedicated education campaigns [35].

Other institutions have also identified organization barriers to successful ASP initiatives. Specifically, a lack of allocated time and resources limits the impact of the ASP [33,34,36,37] and buy-in from leadership is necessary for success [37]. Hierarchy within the medical team, particularly relevant in academic medical centers, can also be a barrier to success. Junior doctors and trainees often do not feel empowered to counter the prescribing practices of their senior colleagues, which can be at odds with recommendations from the ASP [33,38]. Despite the barriers identified by front-line staff in our study, which comport with those identified by previous investigators, we implemented a fluoroquinolone restriction intervention with great success among the highest-risk patients in our institution.

Our study has several limitations. First, our small, single-site sample limits generalizability though we feel is reflective of critically ill and SOT patients seen at many tertiary academic medical centers. The uncontrolled nature of the unit-level data used to compare outcomes limits our ability isolate the effect of the fluoroquinolone restriction on HO-CDI rates. Regarding secondary outcomes, our study was designed to query readmissions within our hospital system and patients admitted to other systems may have led to undercounting of readmissions. It is possible that inability to adjust for patient level factors affected our results. There were several concurrent HO-CDI control measures implemented during the study period (Table 1), further complicating our ability to make causal inferences. By conducting time series analysis we attempted to isolate the influence of fluoroquinolone restriction, however, it is likely the influence of concurrent interventions remains and cannot be fully controlled for. As our institution offers only *C. difficile* detection by PCR, we may be detecting both colonization and infection; however, the testing methodology did not change in the pre- and post-intervention periods. We did not measure fluoroquinolone susceptibility of *C. difficile* and did not aggregate data based on NAP1 positivity, and it is likely that fluoroquinolone restriction would have greater impact in the setting of known fluoroquinolone resistant *C. difficile*. Antimicrobial use data was extracted from electronic medical record data; we performed iterative validation to minimize inaccuracies in the data. In the validation, we did not find systematic errors that should meaningfully impact our conclusions.

In conclusion, fluoroquinolones can be restricted safely even among high-risk patients without negatively impacting patient-level outcomes but without clear impact on HO-CDI rates. ASP interventions should be informed by barriers and facilitators identified by front-line providers to promote adherence and acceptance. Lessons from our initiative, particularly those learned from exploring the perspectives of front line providers, can be applied to larger-

scale ASP interventions. Future studies should confirm safety and efficacy of restriction policies among critically ill and immunocompromised patients with particular attention to the impact on prescribing of alternative agents and explore other opportunities for optimization of antimicrobial prescribing, such as at the time of hospital discharge.

## Supporting information

**S1 Appendix. Semi-structured interview guide: Fluoroquinolone usage in intensive care and post-transplant care units.**
(DOCX)

**S1 Table. Indications for antimicrobials on study units, one month post-fluoroquinolone restriction implementation, by pharmacy review.**
(DOCX)

## Acknowledgments

The co-authors would also like to acknowledge Liz Godfrey, who queried the electronic medical record for antimicrobial prescribing data, Shelby Tjugum for pharmacy review of prescribing appropriateness and Roger Brown, PhD who provided statistical expertise.

## Author Contributions

**Conceptualization:** Matthew Brunner, Mary Jo Knobloch, Nasia Safdar.

**Data curation:** Lucas Schulz, Anna Barker, Marc-Oliver Wright.

**Formal analysis:** Jessica Tischendorf, Matthew Brunner, Mary Jo Knobloch, Lucas Schulz, Anna Barker.

**Investigation:** Jessica Tischendorf, Matthew Brunner, Alexander Lepak.

**Methodology:** Matthew Brunner, Mary Jo Knobloch, Lucas Schulz, Anna Barker, Alexander Lepak, Nasia Safdar.

**Resources:** Nasia Safdar.

**Software:** Mary Jo Knobloch, Lucas Schulz.

**Supervision:** Alexander Lepak, Nasia Safdar.

**Validation:** Jessica Tischendorf, Matthew Brunner, Lucas Schulz, Anna Barker.

**Writing – original draft:** Jessica Tischendorf, Matthew Brunner, Marc-Oliver Wright, Alexander Lepak.

**Writing – review & editing:** Jessica Tischendorf, Matthew Brunner, Mary Jo Knobloch, Lucas Schulz, Anna Barker, Marc-Oliver Wright, Alexander Lepak, Nasia Safdar.

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
