## [Decision Letter · Decision Letter 0]

24 Jun 2020

PONE-D-20-16019

Evaluation of a successful fluoroquinolone restriction intervention among high-risk patients:  A mixed-methods study

PLOS ONE

Dear Dr. Tischendorf,

Thank you for submitting your manuscript to PLOS ONE. After careful consideration, we feel that it has merit but does not fully meet PLOS ONE’s publication criteria as it currently stands. Therefore, we invite you to submit a revised version of the manuscript that addresses the points raised during the review process.

Reviewers raised several concerns about the description of the statistical analysis conducted. Please address these concerns in your revision. 

We look forward to receiving your revised manuscript.

Kind regards,

Monika Pogorzelska-Maziarz

Academic Editor

PLOS ONE

Journal Requirements:

2. Thank you for stating in your manuscript text "This was a quality improvement initiative and was deemed exempt from institutional review board oversight. We followed SQUIRE 2.0 guidelines in the reporting of our quality improvement study [25]." Please also add this information to your ethics statement on the online submission form.

3. Please provide additional details regarding participant consent. In the ethics statement in the Methods and online submission information, please ensure that you have specified (1) whether consent was obtained, (2) whether consent was informed and (3) what type you obtained (for instance, written or verbal, and if verbal, how it was documented and witnessed). If no consent was obtained, please state whether all data were fully anonymized before you accessed them and/or whether an IRB or ethics committee waived the requirement for informed consent.

4. We note that you refer to a semi-structured interview guide and eTable 1 (Indications for antimicrobials on study units, one month post-fluoroquinolone restriction implementation, by pharmacy review). However, we have not received these documents. Please upload these documents as supplementary files.

Reviewers' comments:

Reviewer's Responses to Questions

**Comments to the Author**

1. Is the manuscript technically sound, and do the data support the conclusions?

Reviewer #1: Yes

Reviewer #2: Yes

Reviewer #3: Yes

2. Has the statistical analysis been performed appropriately and rigorously? 

Reviewer #1: I Don't Know

Reviewer #2: Yes

Reviewer #3: Yes

3. Have the authors made all data underlying the findings in their manuscript fully available?

Reviewer #1: Yes

Reviewer #2: Yes

Reviewer #3: Yes

4. Is the manuscript presented in an intelligible fashion and written in standard English?

Reviewer #1: Yes

Reviewer #2: Yes

Reviewer #3: Yes

5. Review Comments to the Author

Reviewer #1: The study was well conducted - the intervention helped to reduce the usage of Fluoroquinolone significantly, but it was not associated with statistically significant reduction of CDI. The possible reasons as well as limitation of the study were discussed. The restricted use was shown to be safe. However, this type of interventions were shown to work in the past studies. Barriers and facilitators highlighted in respect to the prescriber's behaviour were interesting. More qualitative research needed in this respect. Also, programme should be targeted not at one class but also to achieve overall reduction in antibiotic use for desired impact.

Reviewer #2: Overall: The present study is a mixed-methods assessment of a fluoroquinolone prescription restriction at a large academic medical center. The authors found that while HO-CDI decrease in univariate analysis, the ITS analysis reflected no significant changes. Several interesting themes arose out of the semi-structure interviews including ASP teams as a facilitator and recognition of alternative therapies as a barrier. I believe this really speaks to the value of stewardship programs in influencing change. These data are informative and interesting. I applaud the efforts of the authors in their work.

I have minimal comments as the paper was well written and clear. Figure 1 needs the y-axis label changed to something appropriate instead of a variable with an underscore. Additionally, I would acknowledge missing data/lost to follow up issue and potential for bias of the estimates described for readmissions given patients could have presented elsewhere and systems are not universal in the US. I'm sure a lack of healthcare contact at all (including outpatient) after discharge was likely coded as a lack of readmission though again would be more appropriately considered as missing data. Alternatively, could consider approaching with methodological solutions (eg imputation).

Reviewer #3: Tischendorf and colleagues are commended on their efforts to decrease unnecessary fluoroquinolone use and CDI and present the findings of their efforts. Overall this manuscript adds to the armamentarium of literature supporting the needs to actively restrict use of these antimicrobial agents.

Methods:

- Can you please provide more detail on the interventions. I think I am following that FQs were de-prioritized in guidelines and order sets in EPIC but I may be inferring that incorrectly.

- Was there an consideration for interaction of type of patient (i.e. immune compromised versus critically ill) on the outcome of interest? These patient populations are diverse and it would be of interest to have more granular data on one population versus the other, especially given the consideration of risk of C. difficile colonization and the limitations of PCR only testing.

- The only statistical analysis that was stated was use of two-sample paired t-test, however there are also categorical variables being reported. Additionally, was an assessment of nromality performed for continuous variables? Please add these into the statistical analysis section.

- Additionally, can the authors please describe the type of regression used for interrupted timer series analysis in more detail in the statistical analysis section.

Results:

- Intuitively it may make sense to present the quantitative barriers to FQ as the first results discussed and then move into the results of the intervention. This is merely stylistic.

- Can you add specific numbers for the decrease or increase in antimicrobial use.

Discussion:

- The multiple concurrent infection prevention interventions that occurs, mostly late phase I and into phase II need to be stressed in the limitations more, especially if they are not addressed methodologically.

6. PLOS authors have the option to publish the peer review history of their article (what does this mean?). If published, this will include your full peer review and any attached files.

Reviewer #1: No

Reviewer #2: No

Reviewer #3: No

---

## [Author Response · Author response to Decision Letter 0]

31 Jul 2020

Dear Editor:

We would like to thank our reviewers for a thoughtful critique of our manuscript, “Evaluation of a successful fluoroquinolone restriction intervention among high-risk patients: A mixed-methods study.” Below, we have addressed each reviewer critique, referencing changes to our manuscript where applicable. We hope these revisions are received favorably by reviewers and the editorial staff.

2. Thank you for stating in your manuscript text "This was a quality improvement initiative and was deemed exempt from institutional review board oversight. We followed SQUIRE 2.0 guidelines in the reporting of our quality improvement study [25]." Please also add this information to your ethics statement on the online submission form.

3. Please provide additional details regarding participant consent. In the ethics statement in the Methods and online submission information, please ensure that you have specified (1) whether consent was obtained, (2) whether consent was informed and (3) what type you obtained (for instance, written or verbal, and if verbal, how it was documented and witnessed). If no consent was obtained, please state whether all data were fully anonymized before you accessed them and/or whether an IRB or ethics committee waived the requirement for informed consent.

We have clarified this (page 13, lines 221-223).

“Verbal consent was obtained from interviewees prior to semi-structured interviews. Qualitative analysis was conducted in a de-identified manner, preventing attribution of statements to any particular interviewee.”

4. We note that you refer to a semi-structured interview guide and eTable 1 (Indications for antimicrobials on study units, one month post-fluoroquinolone restriction implementation, by pharmacy review). However, we have not received these documents. Please upload these documents as supplementary files.

These are currently uploaded as “supporting information”. 

Captions are now included on page 32.

Reviewers' comments:

Reviewer's Responses to Questions

Comments to the Author

5. Review Comments to the Author

Reviewer #1: The study was well conducted - the intervention helped to reduce the usage of Fluoroquinolone significantly, but it was not associated with statistically significant reduction of CDI. The possible reasons as well as limitation of the study were discussed. The restricted use was shown to be safe. However, this type of interventions were shown to work in the past studies. Barriers and facilitators highlighted in respect to the prescriber's behaviour were interesting. More qualitative research needed in this respect. Also, programme should be targeted not at one class but also to achieve overall reduction in antibiotic use for desired impact.

We thank the reviewer for their comments. We agree more qualitative research in antimicrobial stewardship is needed to inform acceptance among front line providers and sustainability of interventions. We can apply the knowledge gained from our intervention to larger scale stewardship interventions to impact antimicrobial use at large, as you discuss.

Reviewer #2: Overall: The present study is a mixed-methods assessment of a fluoroquinolone prescription restriction at a large academic medical center. The authors found that while HO-CDI decrease in univariate analysis, the ITS analysis reflected no significant changes. Several interesting themes arose out of the semi-structure interviews including ASP teams as a facilitator and recognition of alternative therapies as a barrier. I believe this really speaks to the value of stewardship programs in influencing change. These data are informative and interesting. I applaud the efforts of the authors in their work.

Thank you for these comments.

I have minimal comments as the paper was well written and clear. Figure 1 needs the y-axis label changed to something appropriate instead of a variable with an underscore. 

The axis has been relabeled, and an updated figure 1 uploaded.

Additionally, I would acknowledge missing data/lost to follow up issue and potential for bias of the estimates described for readmissions given patients could have presented elsewhere and systems are not universal in the US. I'm sure a lack of healthcare contact at all (including outpatient) after discharge was likely coded as a lack of readmission though again would be more appropriately considered as missing data. Alternatively, could consider approaching with methodological solutions (eg imputation).

This is an important critique and we have added this as a clear limitation of our quantitative methodology (see page 24, lines 363-365). The updated language is included here for quick reference:

“Regarding secondary outcomes, our study was designed to query readmissions within our hospital system and patients admitted to other systems may have led to undercounting of readmissions.”

Reviewer #3: Tischendorf and colleagues are commended on their efforts to decrease unnecessary fluoroquinolone use and CDI and present the findings of their efforts. Overall this manuscript adds to the armamentarium of literature supporting the needs to actively restrict use of these antimicrobial agents.

We thank the reviewer and agree with the importance of minimizing unnecessary fluoroquinolone exposure.

Methods:

- Can you please provide more detail on the interventions. I think I am following that FQs were de-prioritized in guidelines and order sets in EPIC but I may be inferring that incorrectly.

Response: Thank you for the opportunity to add further clarity. We have revised the intervention section of our methods and would direct reviewers to pages 7-8, lines 113-149 for updated language.

- Was there an consideration for interaction of type of patient (i.e. immune compromised versus critically ill) on the outcome of interest? These patient populations are diverse and it would be of interest to have more granular data on one population versus the other, especially given the consideration of risk of C. difficile colonization and the limitations of PCR only testing.

Response: We agree having more detailed patient-level data would allow for stronger conclusions on the interaction of antimicrobial use and outcomes of interest. Our study used unit-level data available to our infection control and antimicrobial stewardship programs, which does not allow for controlling of patient level factors. This limitation is addressed on our limitations section, pages 24-25, line 365-366.

- The only statistical analysis that was stated was use of two-sample paired t-test, however there are also categorical variables being reported. 

Response: Statistical analyses were only conducted on continuous variables. The categorical variables to which the reviewer is referring (table 3) are available as descriptors of our population of interest.

Additionally, was an assessment of nromality performed for continuous variables? Please add these into the statistical analysis section.

We thank the reviewers for this comment. We have confirmed data normality and added a description of our assessment of normality to the statistics section of the manuscript, which now reads (page 12, line 204-205):

“Pairwise comparisons between pre and post-intervention mean estimates of all primary and secondary outcomes was performed using the two-sample t-test. An assessment of normality was performed for continuous variables using qq-plots.”

- Additionally, can the authors please describe the type of regression used for interrupted timer series analysis in more detail in the statistical analysis section.

We have updated the manuscript (page 13, line 206-208) with this information:

“Five-point moving averages were used to smooth HO-CDI data. A time series analysis was subsequently conducted using Prais-Winsten regression to account for first-order autocorrelation between measurements.”

Results:

- Intuitively it may make sense to present the quantitative barriers to FQ as the first results discussed and then move into the results of the intervention. This is merely stylistic.

Thank you for this comment. We chose to include the quantitative results first, as the goal of our QI project was to decrease fluoroquinolone use.

- Can you add specific numbers for the decrease or increase in antimicrobial use.

Our measure for antimicrobial use is days of therapy/1000 patient-days, with total use in the pre-intervention and post-intervention period included on page 14, line 241-242.

Discussion:

- The multiple concurrent infection prevention interventions that occurs, mostly late phase I and into phase II need to be stressed in the limitations more, especially if they are not addressed methodologically.

We concur these interventions may have impacted our results. As an attempt to isolate the impact of the fluoroquinolone restriction, we selected time-series analysis, however, we do recognize there may be some residual confounding. We have added language to emphasize this possibility in our limitations, which now reads as such (page 24, line 366-370):

“There were several concurrent HO-CDI control measures implemented during the study period (Table 1), further complicating our ability to make causal inferences. By conducting time series analysis we attempted to isolate the influence of fluoroquinolone restriction, however, it is likely the influence of concurrent interventions remains and cannot be fully controlled for.”

On behalf of my collaborators, I would like to thank you for your ongoing consideration of our work and we hope the revised manuscript is satisfactory for publication.

---

## [Editor Report · Decision Letter 1]

7 Aug 2020

Evaluation of a successful fluoroquinolone restriction intervention among high-risk patients:  A mixed-methods study

PONE-D-20-16019R1

Dear Dr. Tischendorf, 

We’re pleased to inform you that your manuscript has been judged scientifically suitable for publication and will be formally accepted for publication once it meets all outstanding technical requirements.

Kind regards,

Monika Pogorzelska-Maziarz

Academic Editor

PLOS ONE
---

## [Editor Report · Acceptance letter]

14 Aug 2020

PONE-D-20-16019R1 

Evaluation of a successful fluoroquinolone restriction intervention among high-risk patients:  A mixed-methods study 

Dear Dr. Tischendorf:

I'm pleased to inform you that your manuscript has been deemed suitable for publication in PLOS ONE. Congratulations! Your manuscript is now with our production department. 

Kind regards, 

on behalf of

Dr. Monika Pogorzelska-Maziarz 

Academic Editor

PLOS ONE